# MiDAS—Meaningful Immunogenetic Data at Scale

Maciej Migdal[1], Dan Fu Ruan[2], William F. Forrest[3], Amir Horowitz[2], Christian Hammer[4,5]*

1 Roche Global IT Solution Centre (RGITSC), Warsaw, Poland, 2 Department of Oncological Sciences, Precision Immunology Institute, Tisch Cancer Institute, Icahn School of Medicine at Mount Sinai, New York, New York, United States of America, 3 Department of OMNI Bioinformatics, Genentech, South San Francisco, California, United States of America, 4 Department of Cancer Immunology, Genentech, South San Francisco, California, United States of America, 5 Department of Human Genetics, Genentech, South San Francisco, California, United States of America

* hammer.christian@gene.com

## Abstract

Human immunogenetic variation in the form of HLA and KIR types has been shown to be strongly associated with a multitude of immune-related phenotypes. However, association studies involving immunogenetic loci most commonly involve simple analyses of classical HLA allelic diversity, resulting in limitations regarding the interpretability and reproducibility of results. We here present MiDAS, a comprehensive R package for immunogenetic data transformation and statistical analysis. MiDAS recodes input data in the form of HLA alleles and KIR types into biologically meaningful variables, allowing HLA amino acid fine mapping, analyses of HLA evolutionary divergence as well as experimentally validated HLA-KIR interactions. Further, MiDAS enables comprehensive statistical association analysis workflows with phenotypes of diverse measurement scales. MiDAS thus closes the gap between the inference of immunogenetic variation and its efficient utilization to make relevant discoveries related to immune and disease biology. It is freely available under a MIT license.

## Author summary

Genetic association studies of complex traits often yield highly significant associations in genetic loci coding for HLA or KIR genes, which have central functions for immune responses. Although the roles of these genes, for example in antigen presentation or immune signaling, are well established, their extreme degree of variability makes it challenging to infer mechanistic hypotheses. Starting with HLA or KIR typing data, our software tool MiDAS facilitates statistical association testing, but also recodes and groups immunogenetic information according to function and validated biological interactions. For instance, we can test for association on the level of the actual amino acid sequence, investigating whether an association is strongest for amino acids that determine whether or not a given antigen can be presented by a HLA protein. We can also group HLA alleles according to their interaction with specific KIR on Natural Killer (NK) cells, and a significant association of such interactions might implicate NK cells in our phenotype of

**Data Availability Statement:** All data are available on https://github.com/Genentech/MiDAS.

**Funding:** The author(s) received no specific funding for this work.

**Competing interests:** I have read the journal's policy and the authors of this manuscript have the following competing interests: MM, WFF, and CH are employees of Roche / Genentech.

interest. In summary, MiDAS offers straightforward workflows for the analysis of immunogenetic data from discovery to functional fine-mapping.

This is a *PLOS Computational Biology* Software paper.

## Introduction

The major histocompatibility complex (MHC) is the region in the genome with the highest density of statistical associations with disease phenotypes. The majority of these associations are related to the central role of classical Human Leukocyte Antigen (HLA) proteins in immune responses in the context of autoimmunity, infectious disease, and also cancer.[1] The underlying cause of these associations can be the presentation of disease-relevant antigens by specific HLA variants, but other mechanisms have been described, such as alternate docking of T cell receptors or differences in the stability of HLA proteins.[1] Another complex genomic locus relevant for immune responses is the leukocyte receptor complex (LRC) on chromosome 19, which, among other genes, harbors the killer cell immunoglobulin like receptors (KIR). KIR predominantly mediate function and education of Natural Killer (NK) cells, but can also be found on subsets of T cells.[2] They display a high degree of copy number as well as allelic variation. Many KIR are receptors for HLA class I ligands via highly specific interactions that depend on individuals' HLA and KIR genotypes, segregating on different chromosomes.[2]

The extreme amount of genetic variation has made it challenging to accurately characterize individuals' HLA and KIR genotypes, but besides dedicated typing methods, there are now multiple tools available for inference from next generation sequencing or single nucleotide polymorphism (SNP) array genotyping data at scale.[3–5] However, the availability of immunogenetic variation data is only the first necessary step in uncovering and understanding the role of HLA and KIR in immune-related traits, and statistical considerations are more complex when compared to the millions of common single nucleotide polymorphisms (SNPs) or copy number variants (CNVs) in our genomes that predominantly have two allelic states.

Genome-wide association studies (GWAS) with significant hits in the MHC region can be complemented with dedicated HLA analyses for statistical fine-mapping purposes. Due to the complex linkage disequilibrium (LD) in the MHC, top associated SNPs can tag one or more alleles of a given HLA gene, without being located within the boundaries of that gene themselves. In a recent GWAS focusing on immune responses to infections, many genome-wide associated SNPs were found in the MHC, and so the authors followed up with HLA imputation and statistical analyses to identify the alleles causing these associations.[6]

HLA alleles at 2-field resolution (formerly '4-digit') are defined by differences in their protein structure (one or many amino acids), resulting in very similar or very different antigen presentation profiles. Therefore, it can be useful to analyze amino acid positions in the peptide binding regions that have the same residue for one group of alleles, but a different one for others. This was shown for example in rheumatoid arthritis, where five amino acids across three HLA genes were found to explain most of the association signal in the MHC locus.[7] HLA alleles can also be grouped together as 'supertypes', based on overlaps in their antigen binding spectrum,[8] a concept that was used to identify HLA risk factors for outcome in dengue fever cases.[9] When NK cells are hypothesized to play a role in a phenotype of interest, it can be useful to consider both HLA and KIR variation and analyze them according to biologically validated interactions, as for example shown in studies focusing on pregnancy complications.[10]

## Design and implementation

Statistical association analyses of immunogenetic variants often focus on carrier status for specific HLA alleles. They are most often analyzed on 2-field level, which defines the protein structure of the HLA protein, as well as the composition of its peptide binding groove and thus the repertoire of antigens it can present. HLA alleles can also be grouped on 1-field level, which often corresponds to the serological antigen carried by an allotype,[11] or on the level of supertypes, which present overlapping peptide repertoires based on their main anchor specificities.[8] In addition, typing data and resulting association statistics can be available on the level of G groups, which contain alleles that have identical nucleotide sequences across the exons encoding the peptide binding domains (exons 2 and 3 for HLA class I and exon 2 for HLA class II alleles).[12]

MiDAS accepts HLA genetic data in tabular form in up to 4-field (8-digit) resolution (one individual per row, two alleles per gene in columns), checks it for consistency with official HLA nomenclature,[11] and can reduce its resolution or transform it into supertypes or G groups, to allow consistent results reporting and cross-study comparability (Table 1). MiDAS includes a function to test for deviations from Hardy-Weinberg equilibrium (HWE) and provides the option to list HWE P values or directly filter out significant alleles, and it is also possible to quickly compare allele frequencies in input data sets with published frequencies across different populations based on data from a comprehensive online database.[13]

In spite of the vast number of statistical associations in the MHC locus, the complex linkage disequilibrium in the region combined with the proximity of genes with different immune-related or non-immune functions can make it difficult to pinpoint causal genes and variants. [14] However, due to the availability of protein sequences for most known HLA alleles, it is possible to use HLA allele data to generate new variables for each amino acid position in a protein that differs across individuals. [7]

MiDAS facilitates this process by inferring variable amino acid residues for all imported individuals with HLA allele data (Fig 1), based on sequence alignments from the IPD-IMGT/HLA database.[15] It is then possible to perform a likelihood-ratio ('omnibus') test for each variable amino acid position in HLA proteins, determine the effect estimates for all residues at associated positions, and also to map the spectrum of HLA alleles that contain each respective residue (Table 1 and Fig 2).

**Table 1. Overview of MiDAS analysis capabilities.**

| Variable type | MiDAS experiment name | Definition | Reference | Example use case |
|---|---|---|---|---|
| HLA alleles | hla_alleles | HLA allele status at 1- to 4-field resolution | [15] | [1,6] |
| | hla_supertypes | HLA class I alleles grouped into supertypes | [8] | [9] |
| | hla_g_groups | HLA alleles grouped according to identical nucleotide sequence in peptide binding domains | [12] | [27] |
| HLA amino acids | hla_aa | Variable amino acid positions and residues based on HLA allele sequence alignments | [15] | [7] |
| HLA intra-individual diversity | hla_het | Heterozygosity vs. homozygosity of each classical HLA gene | | [16] |
| | hla_divergence | HLA class I evolutionary divergence as measured by Grantham's distance | [28,29] | [18] |
| HLA NK ligand status | hla_NK_ligands | Bw4 / Bw6, C1 / C2 allele group inference based on HLA allele matching table | [2,30] | [20,21] |
| KIR gene presence | kir_genes | Presence or absence of specific KIR genes (binary variable) | [31] | [32] |
| HLA-KIR interactions | hla_kir_interactions | Experimentally verified ligand-receptor interactions between HLA class I and KIR | [2] | [10,22,23] |
| Custom | hla_custom, kir_custom | User-provided dictionaries for custom analyses | | |

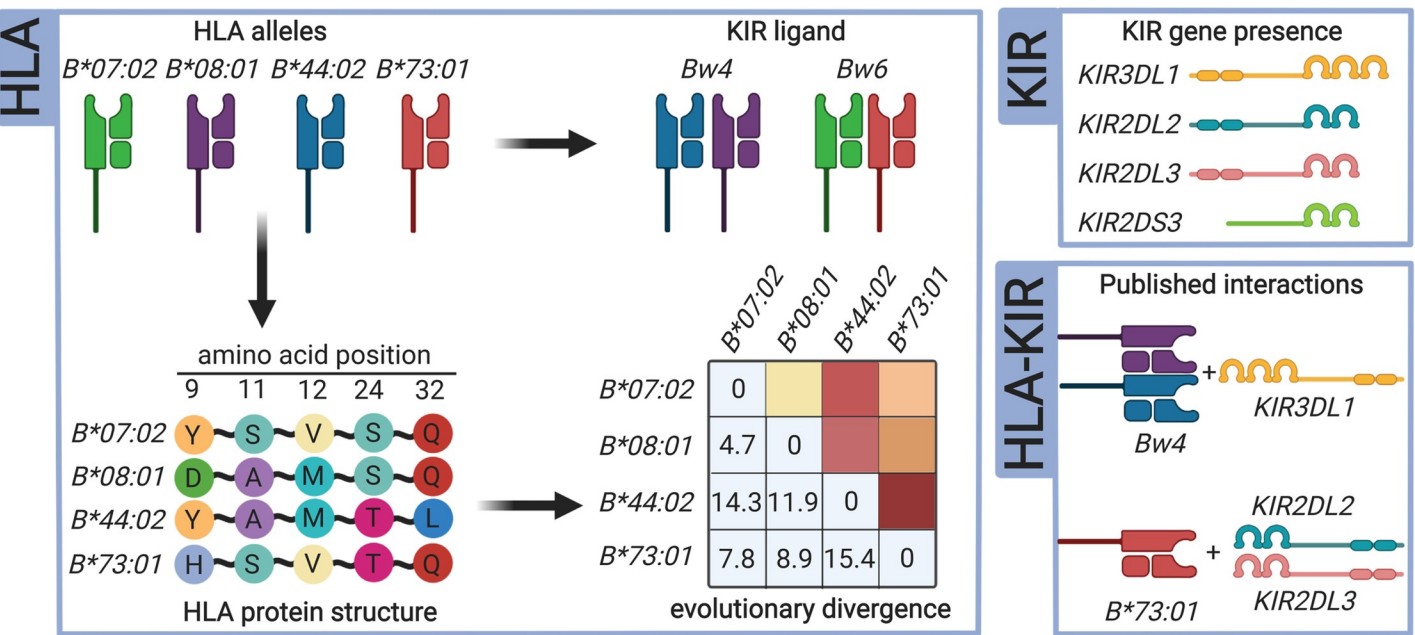

**Fig 1. MiDAS data transformation functions.** MiDAS can transform HLA and KIR input data to test association hypotheses beyond single allele or KIR gene approaches. HLA alleles can be grouped according to their interactions with KIR, and sequence information is used to infer variable amino acid positions for statistical fine-mapping. Amino acid level information is also used to calculate evolutionary divergence of HLA allele pairs for a given gene. If both HLA and KIR data is available, biologically validated receptor-ligand interactions can be coded according to the definitions summarized by Pende et al.[2]

Intra-individual diversity of HLA alleles, assessed in terms of heterozygosity versus homozygosity or evolutionary divergence, is considered a useful proxy for the diversity of antigens that can be presented by an individual's HLA proteins. For example, HIV-positive patients with full heterozygosity for *HLA-A*, *-B* and *-C* were shown to progress more slowly to AIDS,[16] which is likely at least in part due to an increased diversity of presented peptides.[17] Further, cancer patients treated with immune-checkpoint inhibitors responded better to the therapy if they had an increased evolutionary sequence divergence in their HLA class I proteins.[18] MiDAS can recode HLA alleles into new variables indicating heterozygosity at each locus, as well as Grantham's distance for HLA class I genes (Table 1 and Fig 1). Grantham's distance is a method to estimate evolutionary divergence by physicochemical differences between amino acids, and can serve as an estimate for difference in the peptide binding profile of two alleles. It can be calculated for amino acids in the whole peptide binding region of HLA class I molecules, or restricted to the B- or F- binding pockets individually.[19] Both heterozygosity and evolutionary divergence are useful to investigate a possible association of the diversity of antigens presented by an individual's HLA proteins, rather than hypothesizing a role of a specific allele.[18]

Beyond their central role in antigen presentation, HLA class I molecules also function as ligands for KIR, and thus are able to impact NK cell education and function. Beyond interactions between specific HLA alleles and KIR, HLA alleles can also be grouped by MiDAS according to common epitopes into HLA-Bw4, -Bw6, -C1, and -C2 alleles (Fig 1).[2] HLA-Bw4 alleles show experimentally verified interactions with KIR3DL1, whereas HLA-Bw6 alleles have no known interaction with inhibitory KIR. HLA-C1 alleles show strong affinity for KIR2DL3, whereas HLA-C2 alleles show only weak affinity for KIR2DL3, but strong affinity for KIR2DL1. In terms of examples for disease relevance, HLA-Bw4 is a risk factor for ulcerative colitis in Japanese, and homozygosity for HLA-C1 was shown to be associated with reduced risk of relapse in patients with myeloid leukemia after transplantation.[20,21]

Hypotheses including a potential NK cell involvement benefit from the availability of both HLA and KIR typing data. MiDAS can load KIR data indicating the presence or absence of individual KIR genes, and perform association analysis on the level of these genes. But more importantly, if both HLA alleles and KIR data are available, it generates new variables indicating the presence of all experimentally validated interactions as summarized by Pende et al.[2] Investigating the role of such HLA-KIR interactions has previously helped to better understand differential risk for pregnancy complications,[10] pathogen immunity,[22] or NK cell activity in recipients of hematopoietic cell transplants.[23]

Of note, MiDAS also facilitates the testing of specific, more refined hypotheses. For example, amino acid position 80 modulates the interaction of HLA-Bw4 alleles with KIR,[24] which can be modeled by subsetting HLA-Bw4 further according to amino acid level information. Data transformation can also be customized using user-supplied additional data dictionaries. For example, a current shortcoming of MiDAS is that allelic variation of KIR, on top of individual gene presence, is not considered, although it is of demonstrated relevance in modulating interactions between KIR and their respective HLA ligands.[25] Another use case for custom analyses is the transformation of HLA allele data into quantitative variables such as allele-specific expression levels.[26] In both cases, there is still a lot of active research and discussion in the scientific community, and current data dictionaries are likely to become obsolete in the near future. We therefore opted for a custom integration option.

MiDAS allows flexible statistical analyses of immunogenetic data with phenotypes on a diverse range of measurement scales, including regression models or time-to-event data. Results are stored as data tables that display nominal and corrected P values, effect estimates, confidence intervals and variant frequencies. It is possible to execute likelihood-ratio ('omnibus') tests, for example to summarize amino acid residues at each position in the protein and identify the most relevant positions as basis for statistical fine-mapping. MiDAS can also perform stepwise conditional analyses to identify multiple statistically independent association signals within and across HLA genes, which is commonly observed.[7] A range of genetic inheritance models can be selected (where applicable: dominant, recessive, additive, overdominant), as well as the preferred method for multiple testing correction and frequency cutoffs for variable inclusion, taking statistical power considerations into account (Fig 2). Benchmarking tests revealed that MiDAS requires approximately 5GB of memory for datasets with 10,000 observations, and up to 15GB in case of N = 50,000, with runtimes between 50 and 250 minutes for HLA plus amino-acid level analyses. Smaller datasets can be analyzed within seconds to a couple of minutes.

## Methods

### MiDAS data structure

MiDAS accepts HLA types in a format that complies with official HLA nomenclature in up to 4-field resolution (e.g. 'A*02:01:01:01'),[11] one row per individual, and one column for each allele of each gene (e.g. 'A_1', 'A_2'). KIR data is accepted in a tabular format that indicates presence ('1' or 'Y') or absence ('0' or 'N') of each KIR gene. Example input data tables are provided with the package to help putting users' own data in the right format.

The 'prepareMiDAS' function combines HLA, KIR, and phenotypic data into an object that is a subclass of a MultiAssayExperiment, which we termed 'MiDAS'. HLA input data is transformed into counts tables encoding the copy number of specific alleles, as a basis for statistical analysis. This function also offers the described data transformation options (e.g. NK cell ligands, HLA-KIR interactions).

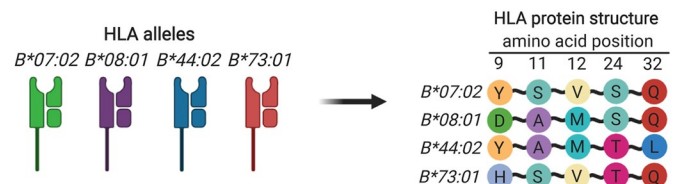

**Prepare analysis**

```
MiDASdat <- prepareMiDAS(hla_calls = HLAdat,
                         colData = clindat,
                         experiment = c("hla_allele",  "hla_aa"))
```

**Run analysis**

```
myModel <- glm(diagnosis ~ term + covar, data=MiDASdat,
family=binomial())

HLA_AA_results  <- runMiDAS(myModel,
                experiment = "hla_aa",
                omnibus = TRUE,
                lower_frequency_cutoff = 0.05,
                upper_frequency_cutoff = 0.95,
                correction = "bonferroni")
```

| MiDAS analysis results | | | | | | |
|---|---|---|---|---|---|---|
| aa_pos | residues | | df | statistic | p.value | p.adjusted |
| DQB1_9 | Y, F, L | | 2 | 21.73 | 1.91E-05 | 8.67E-03 |
| B_97 | R, T, S, N, V, W | | 5 | 33.03 | 3.70E-06 | 1.68E-03 |
| B_81 | A, L | | 1 | 15.86 | 6.83E-05 | 3.09E-02 |
| B_82 | L, R | | 1 | 15.14 | 9.96E-05 | 4.51E-02 |
| B_83 | R, G | | 1 | 15.14 | 9.96E-05 | 4.51E-02 |
| B_62 | R, G | | 1 | 15.03 | 1.06E-04 | 4.79E-02 |

```
HLA_AA_DQB1_9 <- runMiDAS(myModel,
                inheritance_model = "dominant",
                experiment = "hla_aa",
                omnibus_groups_filter = "DQB1_9",
                exponentiate = TRUE,
                correction = "bonferroni")

HLA_AA_DQB1_9_alleles <- getAllelesForAA(HLA_AA,"DQB1_9")
```

| MiDAS analysis results | | | | | | | | |
|---|---|---|---|---|---|---|---|---|
| aa | p.value | p.adjusted | estimate | CI low | CI high | % total | % controls | % disease |
| DQB1_9_F | 6.20E-06 | 1.85E-05 | 1.79 | 1.40 | 2.32 | 15.10% | 11.40% | 18.80% |
| DQB1_9_Y | 1.11E-05 | 3.32E-05 | 0.59 | 0.47 | 0.75 | 82.45% | 86.30% | 78.60% |
| DQB1_9_L | 6.73E-01 | 1.00E+00 | 1.13 | 0.65 | 1.97 | 2.45% | 2.30% | 2.60% |

| HLA-DQB1 (9) | HLA-DQB1 alleles | count | frequency |
|---|---|---|---|
| F | *04:01, 04:02, 04:23, 06:02* | 302 | 15.10% |
| L | *06:01* | 49 | 2.45% |
| | *02:01, 02:02, 02:10, 03:01, 03:02, 03:03,* | | |
| | *03:04, 03:05, 03:19, 03:22, 03:251, 03:96,* | | |
| | *05:01, 05:02, 05:03, 05:04, 05:107, 06:03,* | | |
| Y | *06:04, 06:07, 06:09* | 1649 | 82.45% |

**Fig 2. Example of amino acid fine-mapping analysis.** Example analysis flow for HLA amino acid analysis. In the first step, HLA and clinical data were combined in a MiDAS object using the 'prepareMiDAS' function, which also performed HLA data transformation to amino acid level (specified as 'experiment'). Before the association analysis, a statistical model was defined. 'term' is a placeholder that is replaced by each tested amino acid, covariates ('covar') can be categorical or numeric. It is also possible to define interaction terms (e.g. 'term:covar', not shown). 'runMiDAS' was then run twice, first to perform an omnibus test on all variable amino acid positions, and then to calculate effect estimates for all residues (F,Y,L) at the top-associated position (DQB1_9). 'getAllelesforAA' was then used to map all *HLA-DQB1* alleles in the dataset to the three DQB1_9 residues.

Compared to the MultiAssayExperiment, 'MiDAS' class makes several assumptions that allow us to use data directly as an input to statistical model functions. The most important assumptions are: the variable names are unique across MiDAS, each experiment has only one assay defined. Further, experiments are defined as matrices or SummarizedExperiment objects. The latter is used in cases where experiment specific metadata are needed for the analysis, including variable groupings used for omnibus tests, or information on the applicability of inheritance models.

## Statistical framework

The data analysis framework offered by MiDAS is flexible in terms of choice of the statistical model, often used examples including logistic or linear regression, or cox proportional hazard models for time-to-event analyses. This flexibility is possible due to using 'tidyers' as introduced in the 'broom' package (https://broom.tidymodels.org).

The MiDAS object is passed as a data argument to the function, and the chosen genetic variables are provided using a placeholder variable ('term'). The defined model is passed to the 'runMiDAS' function, where the actual statistical analysis is performed. Here, the placeholder

variable is substituted with the actual genetic variables in an iterative manner, allowing to test individual variables for association with the response variable. The use of a placeholder allows the use of more complex statistical models, e.g. gene-environment interactions (e.g. "lm(diagnosis ~ 'term' + sex + 'term':sex)".

'runMiDAS' offers different modes of analysis. By default, the statistical model is iteratively evaluated with each individual genetic variable substituted for the placeholder. Then, test statistics from individual tests are gathered and corrected for multiple testing using a method of choice as implemented in the 'stats' R package. Moreover, 'runMiDAS' offers a conditional mode to test for statistically independent associations of multiple genetic variables, which implements a simple stepwise forward selection method. Here, the iterative comparisons are made in rounds, and for each round the algorithm selects the top associated variable and adds it to the model as a covariate, until no more variables meet the selection criteria. Further, 'runMiDAS' includes an 'omnibus' mode that allows to test the role of multiple grouped variables, using a likelihood ratio test. In particular, this is useful to score amino acid positions according to their omnibus P value, as compared to their individual residues.

Of note, MiDAS does not offer novel statistical approaches for the analysis of immunogenetic data, and the statistical model is chosen by the user. Therefore, results will be identical to a more labor-intensive approach involving step-by-step testing of variables of interest.

## Availability and future directions

MiDAS is freely available as an R package (MIT license), facilitating both hypothesis-driven and exploratory analyses of immunogenetics data (https://github.com/Genentech/MiDAS). A tutorial with example data and analyses is available under https://genentech.github.io/MiDAS/articles/MiDAS_tutorial.html. Future versions will include the possibility to work with more granular KIR genotyping data (copy number and allelic variation). The inference of HLA haplotypes from allele data would also be a useful feature that was considered out of scope for the current version. Users are welcome to contribute to the future development of MiDAS.

## Author Contributions

**Conceptualization:** William F. Forrest, Amir Horowitz, Christian Hammer.

**Data curation:** Maciej Migdal.

**Formal analysis:** Maciej Migdal, Dan Fu Ruan, Christian Hammer.

**Investigation:** William F. Forrest, Christian Hammer.

**Methodology:** Maciej Migdal, Dan Fu Ruan, William F. Forrest, Amir Horowitz, Christian Hammer.

**Project administration:** Christian Hammer.

**Resources:** Christian Hammer.

**Software:** Maciej Migdal, Christian Hammer.

**Supervision:** Amir Horowitz, Christian Hammer.

**Visualization:** Christian Hammer.

**Writing – original draft:** Maciej Migdal, Christian Hammer.

**Writing – review & editing:** Christian Hammer.

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
