## [Decision Letter · Decision Letter 0]

7 Mar 2021

Dear Dr. Hammer,

Thank you very much for submitting your manuscript "MiDAS - Meaningful Immunogenetic Data at Scale" for consideration at PLOS Computational Biology.

As with all papers reviewed by the journal, your manuscript was reviewed by members of the editorial board and by several independent reviewers. In light of the reviews (below this email), we would like to invite the resubmission of a significantly-revised version that takes into account the reviewers' comments.

We cannot make any decision about publication until we have seen the revised manuscript and your response to the reviewers' comments. Your revised manuscript is also likely to be sent to reviewers for further evaluation.

Sincerely,

Mihaela Pertea

Software Editor

PLOS Computational Biology

Mihaela Pertea

Software Editor

PLOS Computational Biology

Reviewer's Responses to Questions

**Comments to the Authors:**

Reviewer #1: Summary: In this work, the authors describe MiDAS a free package in R where HLA alleles and amino acid sequences can be tested for association with a given phenotype. The program is specialized for HLA amino acid fine mapping and evolutionary divergence. In addition, KIR effect on phenotype can be tested by KIR genes in association with present HLA alleles. The program is capable of refining data so that it all data is in the same format for cross-study analysis. MiDAS statistical tests include Hardy-Weinberg equilibrium, linear and logistic regression tests, Cox proportional hazard models and likelihood ratio tests. Overall the paper is well written and the MiDAS package is of general use to the complex trait genetics community, where HLA and KIR association testing is often complex and highly relevant.

Comments:

1) I recommend that a better description of the motivation behind developing MiDAS aimed at the non-expert be included in the introduction or early in the Design and implementation section. For example, the authors state that “statistical considerations are more complex” for HLA and KIR analysis but don’t provide further description. Including this would improve readability and understanding for a general audience.

2) Some clarifications are warranted (a few example below):

On pg 2 the authors state “MiDAS accepts HLA genetic data” but don’t mention the type of data. Does it accept sequence data, textual allele names, SNP data? Is it for a single individual or a population (the same is true for KIR data)? I see now that this is included in methods but I feel it would be helpful to specify this earlier.

Author’s should describe what Grantham’s distance measures and state why it is useful to know for association studies

3) The authors mention the use case of allele-specific expression for HLA? Is this data included in MiDAS? Literature suggests that imputing at least HLA class I expression is possible and biologically informative [for example PMID: 23559252 and 29302013 ]. This would be a nice addition.

Reviewer #2: Migdal et al. introduce an R package to carry out multiple analyses for HLA and KIR loci. Standard tools for analyzing genetic data often cannot deal with conventions used for documenting HLA and KIR variation, or often miss important variation which could help explain disease and normal phenotypes. Even when carrying out genome-wide analyses, HLA and KIR dedicated analyses might be needed to appropriately take into account the distinct characteristics of these loci, and researchers find themselves often in the need to develop custom code and methods to parse and analyze HLA and KIR data. Therefore, a tool to perform such tasks in a consistent way is very welcome and a relevant contribution.

However, I have some comments which I believe can help improve the manuscript.

Overall, I think that the manuscript is too brief, resembling more of a technical note instead of an original research article.

There is no Author Summary section, which I believe should be included for publishing in Plos Comp Bio.

Although the main result here is the computational tool itself, I miss some biological results. When we propose a new tool, it is good practice to show the accuracy in recovering ground truth results from simulated data or in replicating previously reported results. Please consider trying to replicate results from familiar papers (e.g. Arora et al. and Carrington et al. papers that you cite) if their datasets are available, or at least cite published examples of particular analyses to justify why they are important (you’ve done that in the last section of your tutorial).

In that regard, the tutorial document is much more complete than the manuscript. The tutorial shows the potential of the package, it provides more use case examples, and it motivates users to try the package out, more than the manuscript does.

I’d like to see a version with this manuscript with more material and results, which would make clear what the potential of the package is, motivating more users to try it out. Overall I think this work is a promising contribution; I will share it with students, and maybe be a user myself.

Additional comments:

In the introduction, the authors try to say that (1) conventions used for documenting genomic variation are not optimal for HLA and KIR, (2) standard methods to analyze genetic data often miss important variation at HLA and KIR, and (3) genome-wide statistical association methods often miss hits at HLA and KIR, which calls for dedicated methods and analyses. Those are all relevant points which deserve discussion, however the text is not very clear and omits important examples and citations. Please try to improve this discussion, because it is indeed relevant.

For example, this GWAS for COVID-19 (10.1056/NEJMoa2020283) is an example of dedicated analyses for HLA complementing a GWAS, and illustrates a potential use case for your package.

Minor points:

(1) Introduction

“Many KIR are receptors for HLA Class I ligands, but these interactions are highly specific”.

I don’t see a contrast between the 2 statements. Consider something like “Many KIR are receptors for HLA Class I ligands via highly specific interactions that depend on the individuals’ HLA and KIR genotypes”.

(2)

“the availability of immunogenetic variation data is only the first necessary step in uncovering and understanding its role in immune-related traits”

This phrase does not read well since the subject is “availability of immunogenetics variation data”, and the availability of data doesn’t have any roles on traits. Consider “the HLA and KIR roles in immune-related traits”.

(3)

Design and implementation

“Statistical association analyses of immunogenetic variants often focus on the presence vs. absence of single HLA alleles.”

As written, this may be misleading because “presence vs absence” usually refers to loci which show CNV (e.g., DRB3/4 and KIR). I think the authors actually mean that statistical associations for HLA are often carried out at the HLA allele level, which is an important unit of information both for its biological meaning and knowledge accumulated by traditional studies.

(4) The manuscript is too brief when explaining some points. For example, the authors describe the consideration of allelic variation at KIR as a shortcoming of their tool, but do not explain the reason why this is not possible to implement. Further, “Data transformation can also be customized using user-supplied additional data dictionaries”, but no examples are given.

(5) It may be confusing that the package is named “MiDAS”, it is installed as “MiDAS”, but the package is actually “midasHLA” (library(“midasHLA”)). Consider changing this.

(6) In the tutorial, please indicate that some tidyverse packages need to be loaded, otherwise the code fails.

(7) One disappointing aspect of computational tools for HLA is that they usually get stuck with a single and outdated version of internal datasets, such as IMGT data or allelefrequencies.net. Please consider a simple interface for users to update datasets, so your tool can live longer.

Reviewer #3: The authors have presented a paper describing in detail the MIDAS software. The paper address the challenges of analysing HLA data with providing an analysis of the dataset and disease association measures. The following queries arose when reading the manuscript.

The authors describe data formats and options to refine the data to certain levels of resolutions. Experience of real life datasets is that HLA datasets are often not clean and well defined. Many data sets have missing values, some loci may be untyped and there are often mixed resolutions, strings of allele typings and NMDP MAC allele codes. How does the software cope with this data. How are homozygous types represented. Very large datasets are often typed with multiple techniques and analysed against multiple versions of the reference database, leading to inconsistencies, how is this addressed.

Is there a minimum and maximum size of dataset? Two challenges in analysis of this type are the use of software to provide results where the data set is too small for the results to be valid, or the dataset is too large to load.

The HLA-DPB1 and MICA and MICB alleles use a slight variation of HLA nomenclature, how does this cope?

The authors briefly mention comparisons to other applications but more recently published tools like Easy-HLA and the Gene[rate] tools from HLA-net are not mentioned, with which there may be some overlap.

There is no discussion of how the software was tested and validated, which would be informative.

**Have all data underlying the figures and results presented in the manuscript been provided?**

Reviewer #1: Yes

Reviewer #2: Yes

Reviewer #3: Yes

PLOS authors have the option to publish the peer review history of their article (what does this mean?). If published, this will include your full peer review and any attached files.

Reviewer #1: **Yes: **Paul McLaren

Reviewer #2: **Yes: **Vitor R. C. Aguiar

Reviewer #3: No
---

## [Decision Letter · Decision Letter 1]

30 May 2021

Dear Dr. Hammer,

We are pleased to inform you that your manuscript 'MiDAS - Meaningful Immunogenetic Data at Scale' has been provisionally accepted for publication in PLOS Computational Biology.

Best regards,

Mihaela Pertea

Software Editor

PLOS Computational Biology

Mihaela Pertea

Software Editor

PLOS Computational Biology

Reviewer's Responses to Questions

**Comments to the Authors:**

Reviewer #1: I am satisfied with the author's responses to my review. I support publication of the revised version.

Reviewer #2: I am satisfied with the authors' responses and updates. I still believe that a lot of material in the tutorial could be moved to the manuscript, as this would provide a more substantial motivation for specific analyses. However, I respect the authors' decision to keep the manuscript brief, while providing a separate tutorial which will complement the paper. If the editor considers the current format appropriate, I believe this work is a welcome contribution and it deserves publication.

Reviewer #3: The authors have responded positively to the reviewers comments and the Github and tutorial provided work well alongside the manuscript.

The authors have responded to a query regarding the data input formats, it may be of use to include some of their response to reviewers in the main manuscript. HLA data is notoriously complicated and messy (different resolutions, strings and codes) and whilst it may be the responsibility of the user to clean data before using MIDAS, there may be benefit in explicitly stating this, form experience many users expect tools for HLA analysis to also clean the data, as well as perform the expected analysis.

The authors have confirmed that MIDAS is not performing any novel statistical methods, and as such validation is limited, the response to reviewers that

"When writing these functions, we tested them in parallel to this step-by-step approach, thereby making sure results are comparable." neatly sums this query up and may be useful to include in the manuscript but is not essential.

**Have the authors made all data and (if applicable) computational code underlying the findings in their manuscript fully available?**

Reviewer #1: Yes

Reviewer #2: Yes

Reviewer #3: Yes

PLOS authors have the option to publish the peer review history of their article (what does this mean?). If published, this will include your full peer review and any attached files.

Reviewer #1: **Yes: **Paul J McLaren, Ph.D.

Reviewer #2: **Yes: **Vitor R.C. Aguiar

Reviewer #3: No

---

## [Editor Report · Acceptance letter]

1 Jul 2021

PCOMPBIOL-D-21-00070R1 

MiDAS - Meaningful Immunogenetic Data at Scale

Dear Dr Hammer,

I am pleased to inform you that your manuscript has been formally accepted for publication in PLOS Computational Biology. Your manuscript is now with our production department and you will be notified of the publication date in due course.

With kind regards,

Olena Szabo
